# Water Influence on the Uniaxial Tensile Behavior of Polytetrafluoroethylene-Coated Glass Fiber Fabric

**DOI:** 10.3390/ma14040846

**Published:** 2021-02-10

**Authors:** Hastia Asadi, Joerg Uhlemann, Natalie Stranghoener, Mathias Ulbricht

**Affiliations:** 1Institute for Metal and Lightweight Structures, University of Duisburg-Essen, Universitaetsstr. 15, 45141 Essen, Germany; joerg.uhlemann@uni-due.de (J.U.); natalie.stranghoener@uni-due.de (N.S.); 2Lehrstuhl für Technische Chemie II, University of Duisburg-Essen, Universitaetsstr. 7, 45117 Essen, Germany; mathias.ulbricht@uni-essen.de

**Keywords:** polytetrafluoroethylene (PTFE)-coated glass fiber fabric, uniaxial tensile strength and stiffness, water diffusion and degradation mechanisms, in-plane and out-of-plane water seepage

## Abstract

Polytetrafluoroethylene (PTFE)-coated glass fiber fabrics are used for long-lasting membrane structures due to their outstanding mechanical properties, chemical stabilities, and satisfying service life. During their operation time, different environmental impacts might influence their performance, especially regarding the mechanical properties. In this contribution, the impact of water on the tensile strength deterioration was assessed experimentally, providing evidence of considerable but partially reversible loss of strength by up to 20% among the various types of investigated industrially established fabrics.

## 1. Introduction

### 1.1. General

Water is one of the ubiquitous environmental substances in the form of air humidity, rain, snow, hail, and dew. In its various forms, it can attack all outdoor materials by two action mechanisms: (1) mechanical stress arising from sequences of swelling and deswelling as a result of water content fluctuation in the environment or from freezing and thawing, or (2) chemical reactions of water with the material, such as hydrolysis [1]. Furthermore, water inside the layers of coated woven fabrics can provide a moist environment for fungus and mildew growth, which might affect the coating system and also reduce the membrane performance (both aesthetic and strength performances). Two common architectural coated woven fabrics are PET (polyethylenterephthalat)-PVC (polyvinyl chloride) and glass-PTFE (polytetrafluorethylene) fabrics. PTFE-coated glass fabrics (see Figure 1) have extraordinary advantages such as high tensile strength of the glass fibers, high dimensional stability (little stress relaxation), high stiffness, and nonflammability. The PTFE coating is resistant to environmental influences, chemical attacks, and mold growth, and fluoropolymer top coatings can be applied to improve the self-cleaning characteristics and weldability [2,3]. The detailed features of different layers of glass-PTFE are described in Table 1. However, the brittle behavior of glass fibers requires careful handling to avoid folding due to acute angles [4].

By assuming ideal functionality for different layers of glass-PTFE fabric, this composite material can be regarded as one of the most versatile and durable architectural membrane materials. The life expectancy of glass-PTFE is more than 30 years. However, the ideal functionality has not been questioned or proven by research up to now. To the authors’ knowledge, there is no report that connects the premature failure of glass-PTFE structures to water attacks. In this way, engineers today rely on the perfect water tightness of PTFE-coating to protect glass fibers from the destructive effects of water. Consequently, these effects up to now have not been taken into account in the engineering of membrane structures made from glass-PTFE fabrics. The new results of the present investigations indicate this to be a potential risk to the safety of such structures.

This study surveys the water-tightness performance of the coating layer by assessing the tensile strength changes of glass-PTFE material under various forms of water attack. Firstly, an overview is provided on mechanisms of water diffusion and water-induced degradation in each layer of a glass-PTFE composite. Secondly, as the main goal, water impacts on the mechanical properties, i.e., tensile strength and global stress–strain behavior, are scrutinized experimentally. Finally, a method of handling observing strength reductions during engineering is proposed. It takes up the idea of weathering-induced ageing strength modification factors, which was introduced in the recently developed prCEN/TS 19102:2020-10 [17].

### 1.2. Water Seepage in Architectural Coated Membrane Fabric

The transmission of liquid in-plane and perpendicular to the plane of the fabric is called in-plane and out-of-plane wicking (flow by capillary action), respectively. Membrane structures made of architectural fabrics are usually considerably inclined, but recently there has been a significant move toward flatter forms [18]. The latter increases the potential of in-plane wicking via the seams and uncovered edges as well as out-of-plane wicking by water seepage through the thickness. Additionally, pinholes might appear at the point of the lowest coating thickness (see Figure 2), especially in heavyweight plain weave fabrics [4,7].

It cannot be excluded that water can penetrate through these pinholes and can even expand their diameters. Figure 3 shows the presence of pinholes on the surface of virgin glass-PTFE fabrics.

### 1.3. Stress–Strain Behavior of Glass-PTFE Fabrics

Generally, stress–strain curves of uniaxially stressed fabrics exhibit three specific regions, including inter-fiber friction, decrimping, and yarn extension regions. The first region does not appear in glass-PTFE fabrics, possibly because of the anti-abrasive effect of the size coating of the glass filaments. Due to this, no remarkable friction appears between the filaments of yarns. In the second region, yarns unbend in the loading direction while the crimp increases in the transverse direction for the transverse yarns. Finally, in the third region, the slope of the stress–strain path rises very steeply when the crimp of the threads aligned with the force cannot decrease anymore [19], approaching the stiffness and linear stress–strain behavior of a structural glass element with the same cross-sectional area (see Figure 4). Here, the uncrimping region of the weft direction is far wider than that of the warp direction, which is consistent with weaving patterns containing warp yarns straighter than weft yarns.

### 1.4. Overview on Water Impact on Glass-PTFE Fabrics

Three methods of water absorption in glass fiber reinforced polymer (GFRP) exist: water molecule diffusion into the matrices, fiber/matrix capillary effect, and water seepage by voids and micro-cracks [20].

It is well known that the presence of water reduces the tensile strength of glass yarns as the main load-carrying element of a glass-PTFE composite. Nevertheless, the exact responsible mechanism of tensile strength degradation of glass is still not known and has been a field of research until present. Water can enter and diffuse through glass as a water molecule (H_2_O) and hydroxyl ion (OH^−^) or hydronium ion (H_3_O^+^) [21]. In 1921, Griffith [22] identified the inevitable presence of small flaws in glass, which were interpreted as surface cracks. These cracks cause a delayed failure, which means they spread in time even under moderate stress and they are responsible for the local acceleration of corrosive environmental influences, especially moisture. This is called static fatigue [23]. Michalske and Bunker [24] categorized three main reactions between the glass and aqueous solution: (1) hydration (penetration of molecular water into the glass network as an intact solvent), (2) hydrolysis, and (3) ion exchange (the replacement of a modifier cation such as sodium by hydrogen (H^+^) or a hydronium ion (H_3_O^+^)). Michalske and Bunker believed that these reactions are slow at ambient conditions and need tens of years of exposure to moisture for significant surface alteration. But the reactions accelerate by temperature and the stress increases under extreme acidic or basic conditions. Besides this, if three reactions happen together, each reaction influences the kinetics and mechanisms of the other reactions. In pure silica glass, only the first two reactions (1) and (2) are responsible, whereas in alkali-containing glasses such as soda-lime, ion exchange reactions occur [25]. However, more recent studies believe that hydrolysis is unlikely to be the stress corrosion reaction [22]. Shelby [26] discussed the possibility that when a crack is formed in silica glass, the surrounding environment, e.g., water, diffuses into the newly formed fracture surface and generates a zone of swelling around the crack tip. The surrounding glass constrains the swollen material from expanding and a zone of compressive stresses is generated at the fracture surface around the crack tip. The ion exchange process is responsible for the development of a compressively stressed surface layer because of the relative volume difference of ions. In soda-lime glass, the H_3_O^+^/Na^+^ exchange leads to compressive stresses because H_3_O^+^ is larger than Na^+^ [27]. These compressive stresses at the crack tip, shielding the external stresses, might lead to an increase of the fracture resistance [28,29,30] stated that the stress-enhanced corrosion rate cannot act any further in sharpening the crack tip when the crack tip curvature radius has reached a lower limit (crack tip blunting = 0.5 nm). In such a situation, when the external stress is very low, the crack walls corrode more rapidly than the tip, leading to a progressive tip blunting effect [31]. Some glass scientists [27]. Ref. [32] tracked evidence of plastic behavior even at room temperature in the strong tensile stress field at crack tips. This plastic flaw causes crack tip blunting as well. Water diffusion into glass networks can change the mechanical properties, as well. According to Ito and Tomozawa [33], swelling causes a reduction of Young’s modulus. The glass strength appears to be proportional to Young’s modulus, therefore, a reduction of Young’s modulus is likely to be the stress corrosion reaction.

In water penetration through fiber-glass reinforced composites, the four following categories should be studied: glass, glass-coupling agent interface, coupling agent-matrix resin interface, and matrix [34]. The influence of water on different layers of glass-PTFE composites is illustrated schematically in Figure 5. This graph illustrates the ideal protection of glass fibers from water attacks, with the help of different chemical compositions. It is believed that the weakest part of coated fabrics is the glass-coupling agent interface, especially in the presence of water. In this situation, water can hydrolyze siloxane bonds (≡Si-O-Si≡), which results in debonding the interface. However, these polysiloxane layers are hydrophobic, making water penetration through matrix resin very slow [34]. The matrix-coupling agent interface is also stable against water attacks because of strong chemical bonds [34]. Finally, PTFE as an exterior layer has a very low surface energy and is therefore hydrophobic; the very strong chemical bonds between fluorine and carbon in PTFE make them very resistant to reactions with other molecules such as water.

However, in practice, the existence of some local defects such as pinholes (see Figure 3) in the final coating and/or incomplete coverage of glass sizing [34] are inevitable, which leads to water contact with glass yarns and subsequent damages.

Glass fiber composite materials are common in civil engineering projects such as rebar in reinforced concrete [35]. The reduction in the mechanical properties of GFRP due to water has been observed by many researchers, e.g., Aldajah et al. [36]. Experimental studies usually simulate the environmental effects of moisture via accelerated ageing through the immersion of GFRP in water. Guzman and Brøndsted determined the mechanical properties and water-diffusion coefficients of GFRP (E-glass fibers with epoxy) after immersing specimens in saltwater for long periods. They observed a maximum tensile strength reduction of 24% for composite specimens and 28% for single glass fibers. They believed that the degradation of glass fibers was a result of stress corrosion, which can also occur in GFRP composites by the degradation of the fiber-matrix interface [37]. Liping et al. [20] investigated the water absorption behavior of glass fiber-reinforced epoxy matrix composites with two fiber volume fractions (34% and 44%). The highest percentage of the tensile strength decrease after 42 days of immersion was 13%.

In 2015, Garcia-Espinel et al. [38] investigated the influence of seawater on polyester, vinyl ester, and epoxy matrix-glass fiber composites to determine the most adequate materials for marine civil engineering applications. After water saturation levels, the mechanical properties of epoxy resins stabilized, which made the researchers confident that epoxy is the most suitable matrix for GFRP in seawater applications.

The influence of water on the mechanical properties of GFRP with resins of polyester, vinyl ester, and epoxy has been studied in several scientific journal publications, whereas only a few studies consider PTFE resins. Table 2 gives a brief overview of them. In all studies, tensile strength deterioration was observed even at room temperature. Hence, the motivation for this study was to close the knowledge gap regarding the scientific understanding of the tensile strength deterioration of glass-PTFE materials under water by focusing on scenarios that allow direct access of water to glass fibers due to a nonideal or imperfect protective function of the coating.

## 2. Experimental Investigations

### 2.1. Materials and Methods

#### 2.1.1. General

Firstly, the presented experiments assessed the tensile strength reduction of glass-PTFE exposed to water. Secondly, the differences between tensile strength reduction of in-plane and out-of-plane watering were investigated. For this, two different conditions were examined: In the first condition, a combination of both watering mechanisms (water seepage through uncovered edges and the coating) was scrutinized whereas in the second condition only out-of-plane seepage was evaluated (water penetration through the coating); see Figure 6.

#### 2.1.2. Materials

As summarized in Table 3, three types of PTFE-coated glass fiber fabrics were investigated: types II, III, and IV, based on the type classification (tabulated by tensile strength values) provided in [41]. This covered the whole range of types of glass-PTFE fabrics used in permanent fabric architecture. For all types, it was possible to test more than one batch. The classification of the fabrics was carried out on the basis of the tensile strengths provided in the product datasheets. Usually the measured tensile strength is higher than that given in the datasheet as a safe approach that supports a trouble-free process for manufacturing. If the measured tensile strength is close to a class limit, it can happen that these materials actually belong to a higher type. This appeared for the type II fabric (see Table 3). All materials belonged to the same producer, a well-known German producer whose products are regularly used for membrane structures.

#### 2.1.3. Methods

The combined watered specimens were standard strips (50 ± 1 × 420 mm^2^) according to EN ISO 1421 (see Figure 7a), whereas out-of-plane watered samples were developed as individually formed samples of bigger pieces of the fabrics (350 × 620 mm^2^ or 450 × 620 mm^2^), based on the technical drawings shown in Figure 7b.

The individually manufactured samples were formed like a boat, or rather, like a tank. The bottom of the tank samples covered the required area for three or five standard strip test specimens to be cut out after water treatment (see the marked rectangles on the specimen shown in Figure 7b). The extra edges (100 mm from each side) were bent up to form walls that could stay higher than the water level with the help of the supporting walls (for lighter fabric types) (see Figure 6) or by tie connections (for heavier fabric types) (see Figure 8). These tank type configurations prevented in-plane water penetration over the unprotected edges and allowed the water to attack through the coating as the only possibility. The risk of local crease fold at the corners of this configuration (especially in heavier types) was present. To prevent this risk, it was important to set the corners far from the middle bottom field, from which the strip test specimens were taken later, as indicated in Figure 5. This tank configuration could be filled with water for one-sided out-of-plane water seepage. Additionally, the filled tank could be placed in a water container for two-sided out-of-plane water penetration.

For combined in- and out-of-plane watered strips, two conditions were investigated: (1) temporary changes of the tensile strength by measuring the tensile strength of the specimens immediately at the end of the water cycle and (2) permanent changes by measuring the tensile strength after one cycle of watering and drying. In addition, to trace the water penetration through the coating thickness, either a small tank was filled with blue ink or a small boat shape was floated in an ink bath, as illustrated in Figure 8a,b. The provisions for wet specimens in EN ISO 1421 were followed, i.e., the water volume was equal to 20 times the total sum of the specimens’ volumes either for the combined water seepage or out-of-plane watered specimens. For the two-sided out-of-plane watered specimens, the water volume on each side was equal to the aforementioned amount, as well. The soap surfactant (as surface tension decreasing agent) volume was 0.1% of the water volume. For one test series water without surfactant was also used.

After finishing the water cycle, samples were rinsed and dried between two sheets of blotting papers. Constant weight was considered as a termination point for dried specimens with one moist-dry cycle. The testing material was conditioned in the laboratory temperature (23 °C ± 2 K) for more than 24 h. The uniaxial tensile behavior of the material was measured by a uniaxial constant rate extension machine (CRE) according to the strip test method of EN ISO 1421 [45]. For this purpose, one set of specimens including three or five samples was taken each in warp and weft direction. The elongation was measured by the traverse travel. Simplified but sufficient for comparisons in the given context, the stiffness (E_secant_) was considered as the slope of a connecting line between the starting strain in a specific load cycle and the maximum strain in the same load cycle of each mean stress-strain curve as depicted in Figure 4.

Furthermore, micrographs were taken using a Keyence digital microscope VHX-70 and scanning electron microscope Zeiss D8M 962 (Oberkochen, Baden-Württmberg, Germany). The 3D images were obtained using a Keyence digital microscope VHX-7000 (Osaka, Japan), which scans the surface irregularities and simulates them in gradient profiles.

## 3. Results and Discussion

### 3.1. In-Plane and Out-of-Plane Watering—Assessment of Temporary Changes

In this test series, the tensile strength of watered specimens with watering periods of 24 h, 48 h, 72 h, 144 h, and 720 h was determined. The general trend of the mean stress–strain curves (mean values of three or five strips) did not differ for different watering periods, e.g., stress-strain curves of glass-PTFE type II. The first batch is shown in Figure 9, and the stress-strain curves of all tested materials are shown in the Appendix A. Generally, these stress-strain curves included decrimping and yarn extension regions. An inter-fiber friction region did not appear significantly in the stress-strain curves of glass-PTFE fabrics, possibly because of the anti-abrasive effect of the finish layer on the glass filaments. Moreover, the uncrimping region of the weft direction was far wider than the uncrimping region of the warp direction, which is consistent with the weaving pattern containing warp yarns that are straighter than the weft yarns. Furthermore, the first regions of the stress-strain curves (decrimping area) showed identical behavior for different watering time periods. This means that the watering time does not play any role in the decrimping process of glass yarns in woven fabrics, which is in contradiction with the findings in the literature [4]. It could be implicitly concluded that the stiffness of PTFE (as one of the barriers for decrimping) does not change by increasing watering periods.

The definition of the stiffness properties of glass-PTFE material is somewhat controversial due to the lack of a standardized material model. In this contribution, the stiffness of both virgin and watered specimens was determined with a secant approach. The stiffness E_secant_ is considered the slope of a connecting line between the starting and maximum (breaking) point of each mean stress–strain curve. According to the results in Figure 10, usually a slightly decreasing trend was observed for the stiffness of the investigated materials after different watering time periods.

Figure 11 depicts the ratios of the tensile strengths of individual watered specimens to their mean values of the tensile strengths at virgin state (evaluated from three or five specimens for each test series). Independent of the watering period, the water attack always led to degradation, albeit in some cases only in low magnitude. The decrease of the tensile strength appeared to happen very fast, even in one day, and longer exposure to water showed only a little further influence. The magnitude of the residual tensile strength is shown in Table 4. The results show a big scatter, yet no general trend to stronger degradation due to longer watering time was recognizable. This happened possibly because the fabric did not imbibe more water after reaching the saturation level. In this way, a longer time period did not change anything.

### 3.2. In-Plane and Out-of-Plane Watering—Assessment of Permanent Changes

Moreover, we explored whether the tensile strength deterioration was reversible by drying the specimens. Hereby, we distinguished between (a) temporary changes of the tensile strength covering the residual strength directly after watering and (b) permanent changes covering the stable residual strength after drying. Figure 12 compares the achieved temporary and permanent changes of the mean values of the tensile strength of glass-PTFE type II, obtained from three or five specimens. In all cases, except for the specimens with 72 h of watering, the tensile strength after drying the watered specimens was higher than the tensile strength directly after watering, meaning in the wet state. This shows that a recovery of the tensile strength occurred from the wet to the dry state; nevertheless, a decrease of the tensile strength was observed between the dry (permanent) state and the virgin state. Table 5 indicates the percentage recovery and permanent residual tensile strength. The permanent residual tensile strengths in both principal directions were almost the same whereas the recovery percentages showed some discrepancies in both directions.

Differences were observed again only beyond the yarn decrimping region by comparing the mean stress–strain curves of the virgin state with those after the watering and drying cycles (see Figure 13). The stress–strain curves of all watering time periods are shown in the Appendix A.

Regarding the recovery behavior of the stiffness after drying, no clear trend was observed (see Figure 14).

### 3.3. Comparison of In-Plane and out of Plane Watering

Figure 15 plots mean values of stress-strain data obtained from tensile tests after water penetration via either coating and uncovered edges or via coating only. It should be noticed that again the mean stress–strain curves of each material were the same in the decrimping region. Changes were only observed beyond the decrimping region. For instance, the stress–strain curves for the first batch of glass-PTFE type II are shown in Figure 15. The stress–strain curves of all tested materials are shown in the Appendix A. Mostly a decrease in the stiffness was observed, but it can hardly be stated that the stiffness was always lower compared to the virgin state. Figure 16 illustrates that the scatter band included also slight increases in the stiffness. Furthermore, for one batch of the glass-PTFE fabric type IV, the influence of the surfactant was investigated. It could be shown that the absence of the surfactant in the water solution did not lead to any significant differences in the results compared to tests with surfactant in the water.

Based on the tensile strength (maximum or breaking stress) recorded for all specimens, a decreasing trend was observed for each single specimen during various watering techniques (see Figure 17). The vertical axis depicts the uniaxial tensile strength of the individually watered specimens related to the respective mean values of the virgin states given from three or five strips for each test series. The magnitude of the tensile strength decrease is summarized in Table 6.

In summary, out-of-plane watering alone (see cluster 4 in Figure 17) degraded the material strength in almost the same magnitude as the combination of both in-plane and out-of-plane watering (see cluster 1 in Figure 15). From this point of view, sealing cut edges either in weathering test specimens or even in real structures had no protective effect. Moreover, from a practical point of view it can be stated that an out-of-plane attack from one side was as damaging as an out-of-plane attack from both sides. It can be concluded that the method of water attack played no role. Despite careful sample preparation, damages to the samples during sample preparation could not be completely ruled out. However, the size planning of the samples ensured a big distance of folds to the measuring zone of the tank-shaped specimen, see Figure 7. The visual inspections made the authors confident that folding is rather unlikely inside specimens. This indicates that the degradation due to water attack is made possible by insufficient intrinsic protective functionality of the multilayer coating under application conditions. As soon as glass-PTFE fabric is exposed to water, degradation has to be expected. For an outdoor constructional material this means that this strength degradation has to be considered in general. By scrutinizing Figure 16, no stringent trend ruled the change of the stiffness due to various water seepage mechanisms. The stiffness change ranged from −20.8% and +3.0% for glass-PTFE type IV, first batch, weft direction, and glass-PTFE type II, second batch, warp direction, respectively.

### 3.4. Assessing the Penetration of Water through the Coating

After confirming the presence of water in glass yarns by their tensile strength deterioration, the depth of this penetration mechanism through the coating was investigated by filling the water tanks with blue ink (see Figure 8a). Additionally, the water seepage by capillary forces in the opposite direction of gravity was evaluated by the test configuration shown in Figure 8b. Here again, the target area was the middle rectangle part of the specimen, which was covered with inked water during the whole test duration (144 h). In order to evaluate how far the inked water penetrated the fabric, micrographs were taken from the yarns. As can be seen in Figure 18, after six days of watering a type IV fabric, blue dots became visible in the coating and filaments in both transverse and longitudinal yarns. This observation provides additional evidence that even for the heavy types of glass-PTFE fabrics, the liquid could reach the base glass fabric due to penetration through the comparably thick coating, even in the opposite direction relative to gravity. There was combined indirect (by mechanical testing) and direct evidence (by staining) of water penetration by capillary forces. This could be explained either by macroscopic defects or an insufficient intrinsic protective functionality of the multilayer coating under application conditions. Relevant water-induced degradation mechanisms for glass-PTFE and the probability of an intrinsic origin of water penetration were discussed in Section 1.2. For controlling this water penetration, it is recommended to diminish the number of pinholes, illustrated in Figure 3, on the surface of the intact coating layers. This could be combined with increasing the number of layers to be applied on the fabric.

### 3.5. Strength Reduction Factor for Consideration of the Deterioration Effect by Humidity

As concluded above, due to the omnipresence of water in its various forms, the degrading effect of it on the constructional material glass-PTFE must be considered in every design situation. In the recently published JRC Science and Policy Report for membrane structures [41], a harmonized view of the Ultimate Limit State verification format was incorporated based on the principles of the Eurocodes. In order to clarify the safety margins of a structure, it is recommended to separately use safety factors and so-called strength modification factors, which are in fact strength reduction factors in this case. The current draft standard prCEN/TS 19102:2020-10 proposes a value of 1.1 as the only weathering-induced ageing modification factor for glass-PTFE materials. In line with this concept, a strength modification factor k_hum_ is proposed to describe the deterioration effect of water or humidity on glass-PTFE fabrics by the following equation:(1)$$k_{\mathrm{hum}}=\frac{f_{k,23,\mathrm{virgin}}}{f_{k,23,\mathrm{watered}}}$$
where f_k,23,virgin_ and f_k,23,watered_ represent the 5% fractile values of short-term tensile strength under room temperature (23 °C) in virgin and watered states. k_hum_ was evaluated using Equation (1) for all investigated fabrics. The maximum values of k_hum_ are given in Table 7 for warp and weft directions depending on the investigated type of glass-PTFE fabric. The maximum value of k_hum_ among all investigated types in warp and weft directions was found to be 1.25, which corresponded to a reduction in tensile strength of 20%. All types of studied fabrics could be conservatively covered by this value. This value is significantly higher than the proposed value in the current draft standard prCEN/TS 19102:2020-10, i.e., the actual strength-reducing effects are currently underestimated. It is strongly recommended to adjust it.

## 4. Conclusions

In this contribution, water diffusion and water-induced degradation mechanisms of architectural glass-PTFE fabrics were surveyed with respect to their different components, which clarified some disputes particularly on the tensile strength decrease of glass fibers under water exposure. Furthermore, the decrease of the uniaxial tensile strength of this material was assessed experimentally. Taking advantage of the presented test results, the following conclusions were obtained:A general trend of a decrease in the tensile strength was dominant for glass-PTFE fabrics that come in contact with water.The decrease in the tensile strength appeared to happen fast within the first 24 h of watering; a longer exposure to water had only a marginal further influence.It was proven that water can attack the glass fibers not only via in-plane watering at unsealed cut edges, but also in almost the same tensile strength-damaging magnitude via out-of-plane watering through the coating. This was found for all investigated types of glass-PTFE fabrics. Water penetration caused by insufficient intrinsic protective functionality of the multilayer coating under application conditions seems likely.With a few exceptions, partial tensile strength recovery was observed by drying cycles.Changes in the stress-strain curve characteristic could only be observed beyond the yarn decrimping region, usually in form of loss of stiffness, but some exceptions also indicated a slight increase in the stiffness.It is highly recommended to consider the decrease in the tensile strength due to water impact in the design of membrane structures made of glass-PTFE fabrics. Water in its different forms (air humidity, condensation, rainwater, melting snow) is always present in the environment and thus, a strength decrease can occur in all design situations. The decrease of the strength can be considered by a general, so-called strength modification factor k_hum_, which acts as a strength reduction factor and was determined as maximum k_hum_ = 1.25. The observed recovering effect due to drying should not be considered because of implementing conservative approaches in structural design. This value was significantly higher than the proposed value in the current draft standard prCEN/TS 19102:2020-10.Furthermore, a change in the stiffness resulting from water seepage was observed in the range of approximately −20.8% and +3.0%. This should be considered in the design process as well.

In the future, further investigations will focus on the detailed description of the chemical processes that lead to the observed behavior of glass-PTFE fabrics under water impact. It is also recommended to investigate the impact of basic and acidic rain on glass-PTFE materials.

## Figures and Tables

**Figure 1 materials-14-00846-f001:**
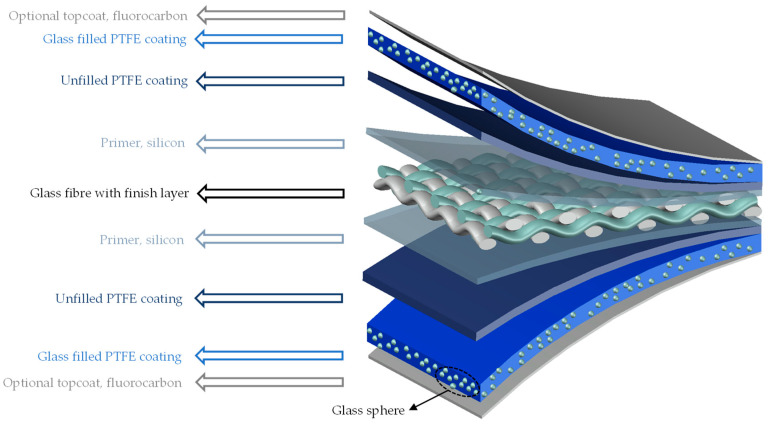
Schematic view of a glass-PTFE composite based on [3].

**Figure 2 materials-14-00846-f002:**
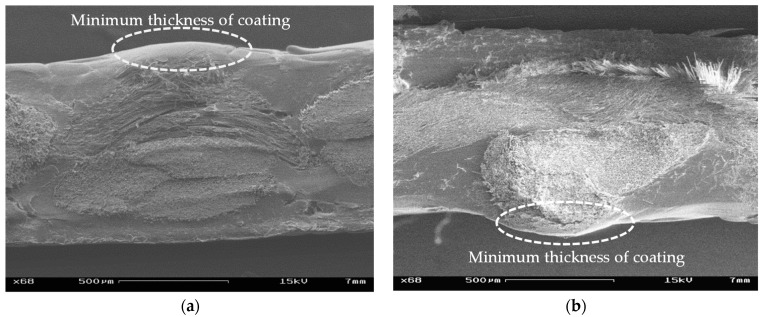
Cross-section view of coating irregularities, glass-PTFE type IV, first batch (see Table 3): (**a**) cut through warp yarns and (**b**) cut through weft yarns; images by scanning electron microscope (see Section 2.1.3).

**Figure 3 materials-14-00846-f003:**
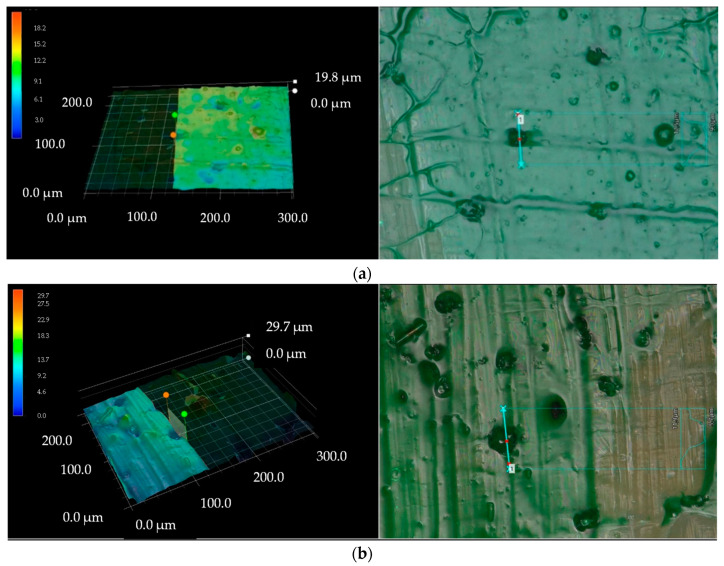
3D images showing the existence of pinholes on the finished surface of glass-PTFE fabrics: (**a**) glass-PTFE type II, first batch (see Table 3) and (**b**) glass-PTFE type IV, first batch (see Table 3); taken by digital optical microscope (see Section 2.1.3).

**Figure 4 materials-14-00846-f004:**
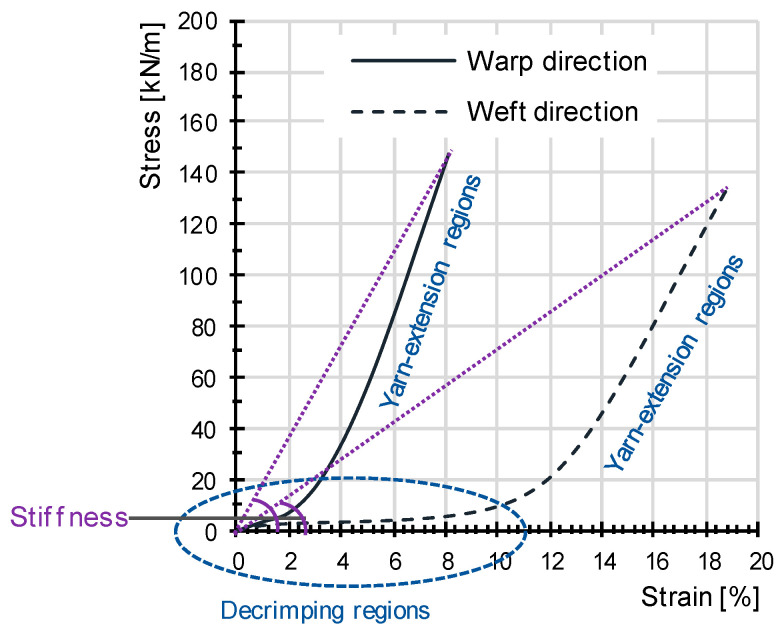
Stress–strain curves of a uniaxial tensile test, glass-PTFE type II, first batch (see Table 3).

**Figure 5 materials-14-00846-f005:**
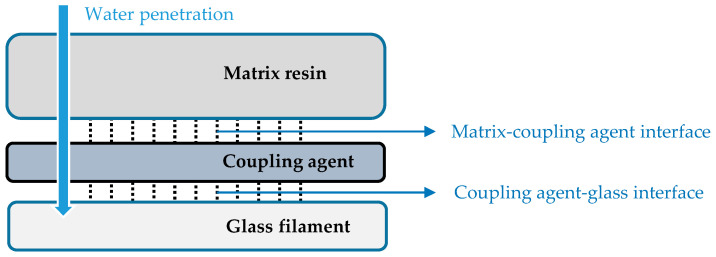
Water impact on various elements of the glass-PTFE fabric.

**Figure 6 materials-14-00846-f006:**
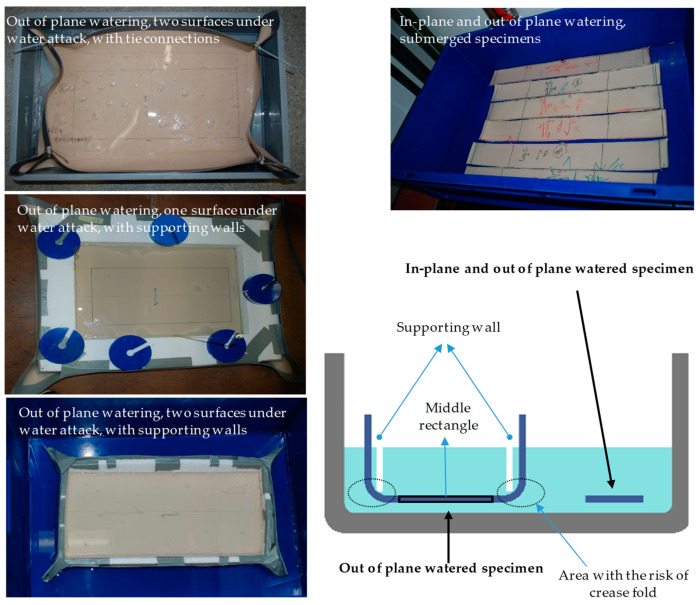
Investigated water seepage mechanisms of glass-PTFE specimens.

**Figure 7 materials-14-00846-f007:**
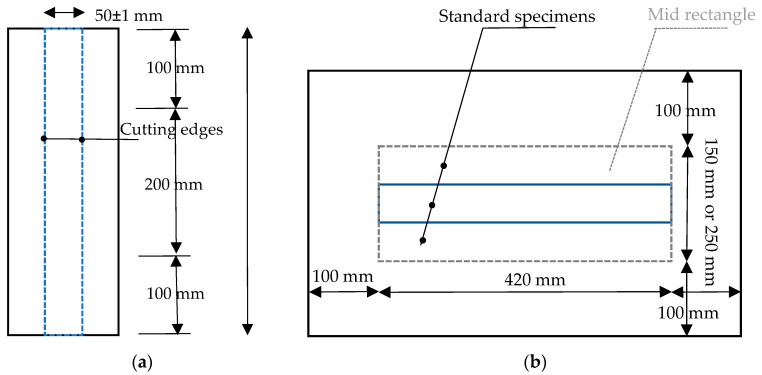
Technical drawings of the used samples: (**a**) sample for combined water seepage and (**b**) sample for out-of-plane water seepage.

**Figure 8 materials-14-00846-f008:**
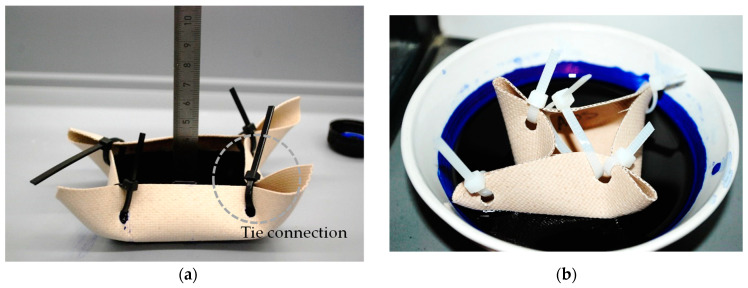
Out-of-plane ink penetration through coating thickness: (**a**) seepage in the direction of gravity and (**b**) seepage in the opposite direction of gravity.

**Figure 9 materials-14-00846-f009:**
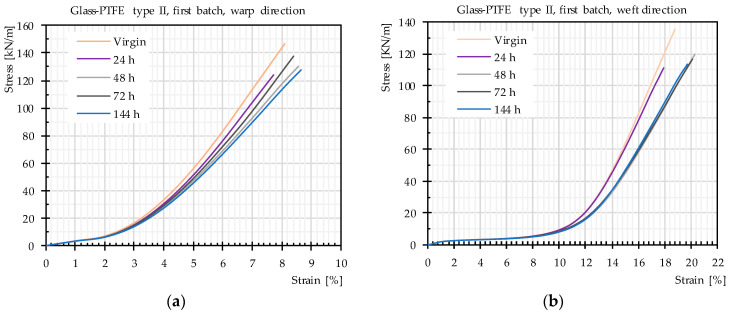
Stress–strain curves of glass-PTFE samples according to Table 3 for in-plane and out-of-plane watering: (**a**) warp direction and (**b**) weft direction.

**Figure 10 materials-14-00846-f010:**
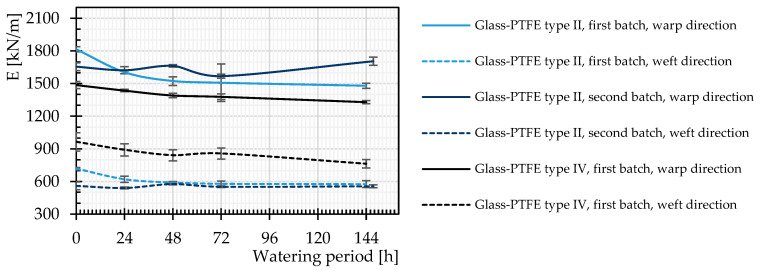
Stiffness changes of glass-PTFE samples according to Table 3 for different watering time periods. Continuous lines: warp direction, dashed lines: weft direction.

**Figure 11 materials-14-00846-f011:**
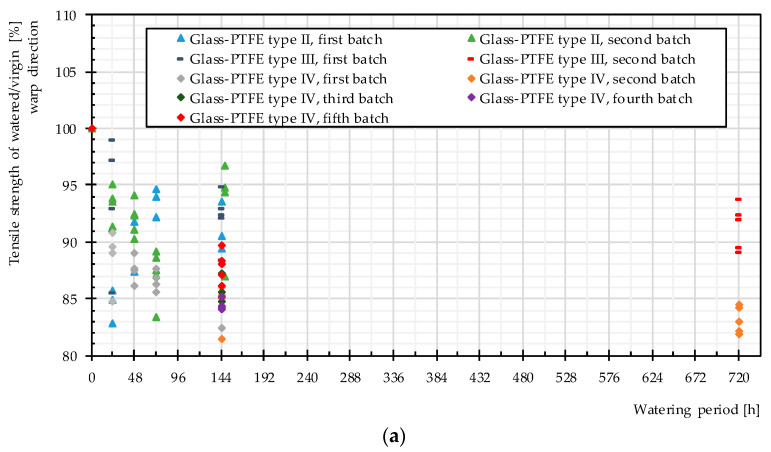
Change of the tensile strength of glass-PTFE samples according to Table 3 at different watering time periods: (**a**) warp direction and (**b**) weft direction.

**Figure 12 materials-14-00846-f012:**
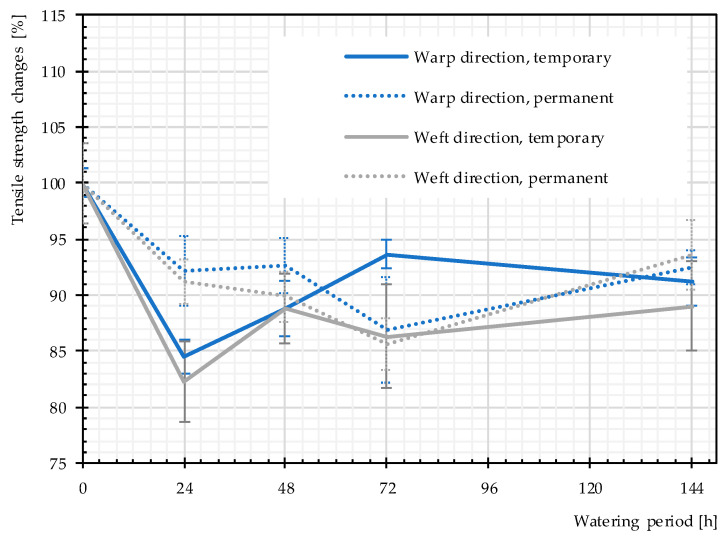
Temporary and permanent changes of the tensile strength, glass-PTFE type II first batch according to Table 3. Continuous lines: temporary changes, dashed lines: permanent changes.

**Figure 13 materials-14-00846-f013:**
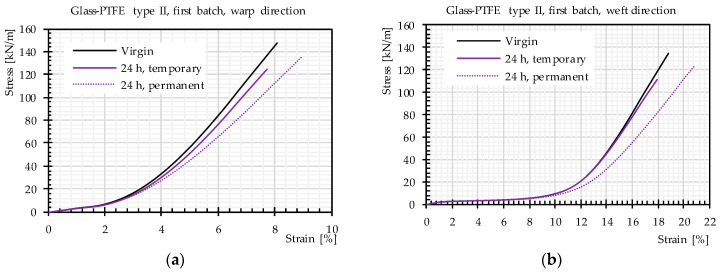
Permanent and temporary stress–strain curves after 24 h of watering, glass-PTFE type II first batch according to Table 3: (**a**) warp direction and (**b**) weft direction. Continuous lines: temporary changes, dashed lines: permanent changes.

**Figure 14 materials-14-00846-f014:**
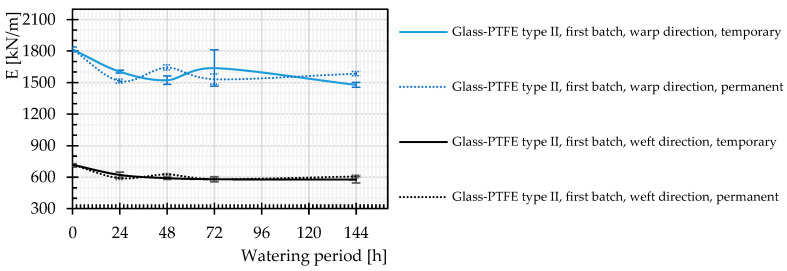
Comparison of stiffness, permanent and temporary changes after different periods of watering, glass-PTFE type II, first batch according to Table 3, continuous lines: temporary changes, and dashed lines: permanent changes.

**Figure 15 materials-14-00846-f015:**
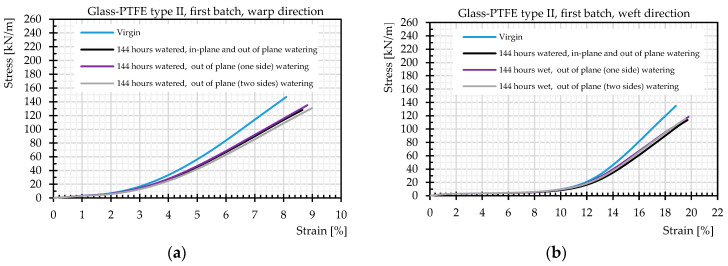
Stress–strain curves of strip- and tank-shaped specimens, glass-PTFE samples according to Table 3.

**Figure 16 materials-14-00846-f016:**
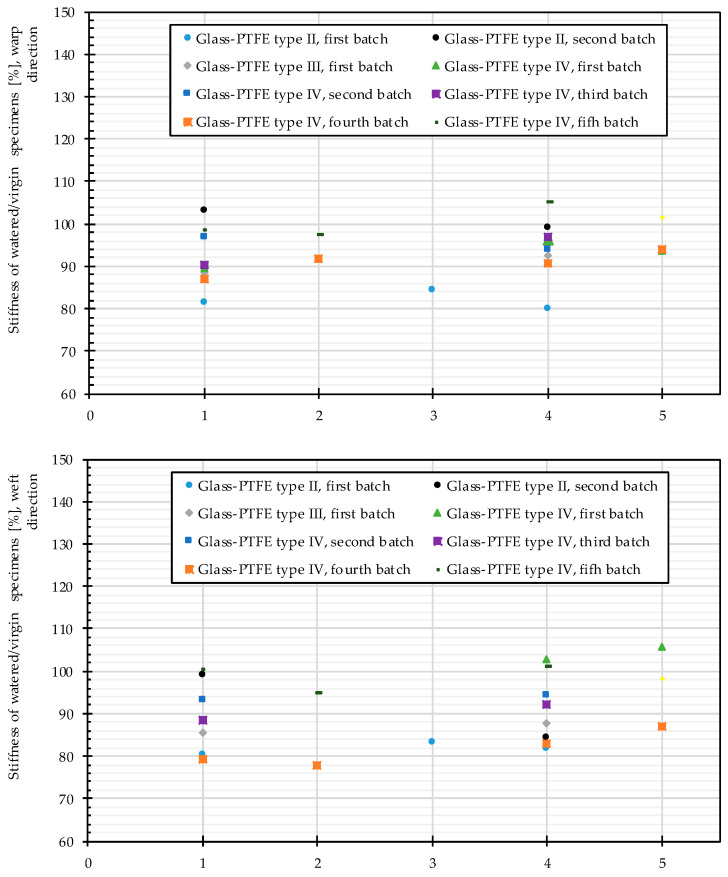
Stiffness changes of glass-PTFE samples according to Table 3 for 1: in-plane and out-of-plane watering (two surfaces), 2: in-plane and out-of-plane watering (two surfaces) without surfactant, 3: out-of-plane watering (one surface), 4: out-of-plane watering (two surfaces), and 5: out-of-plane watering (two surfaces) without surfactant. All are watered for 144 h.

**Figure 17 materials-14-00846-f017:**
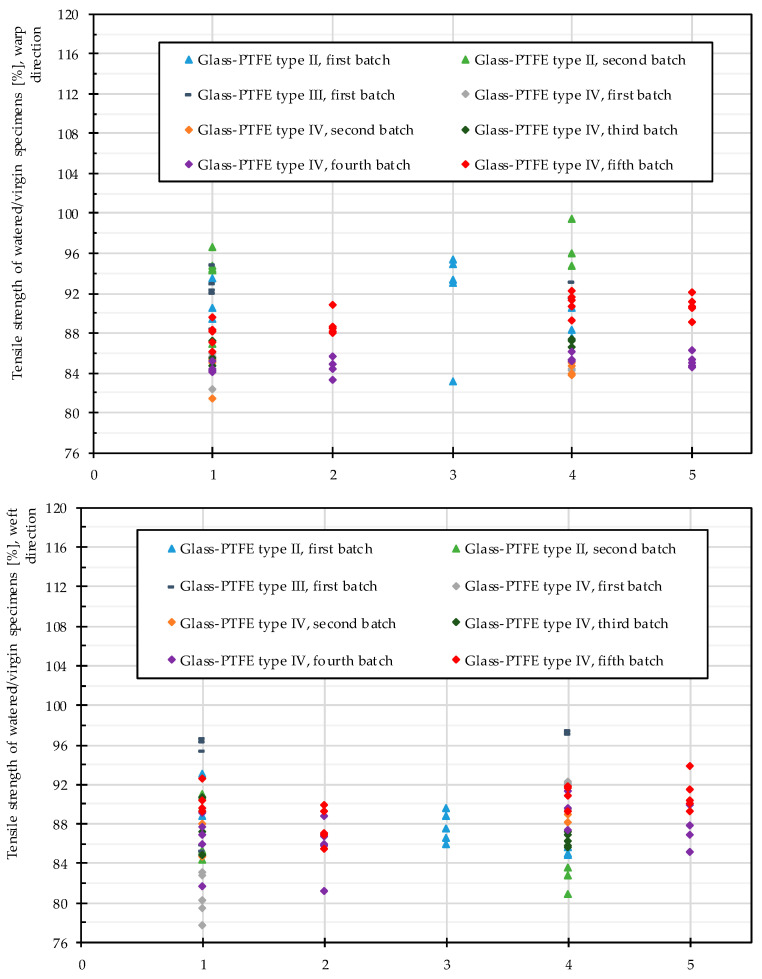
Changes in the tensile strength of glass-PTFE samples according to Table 3 for 1: in-plane and out-of-plane watering (two surfaces), 2: in-plane and out-of-plane watering (two surfaces) without surfactant, 3: out-of-plane watering (one surface), 4: out-of-plane watering (two surfaces), and 5: out-of-plane watering (two surfaces) without surfactant. All are watered for 144 h.

**Figure 18 materials-14-00846-f018:**
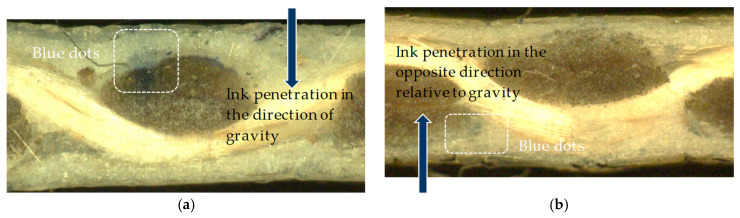
Ink penetration through the coating (blue pigments) after six days of watering, glass-PTFE type IV, the first batch according to Table 3; cross-section view cut through weft yarns: (**a**) ink penetration in the direction of gravity and (**b**) ink penetration in the opposite direction of gravity.

**Table 1 materials-14-00846-t001:** Glass-PTFE fabric: different layers and compositions.

Name of Layer	Feature	Function
E-glass fiber	Inorganic material made of various chemical compositions: SiO_2_, Al_2_O_3_, B_2_O_3_, TiO_2_, MgO, CaO, ZnO, Na_2_O, K_2_O, Fe_2_O_3_, and metal fluorides [5] The existence of metal oxides makes it intrinsically alkaline and hygroscopic, thus facilitating stress corrosion [6] Residual compressive stress outside and tensile stress inside the glass filaments [7,8] because outer layers of molten glass in spinning nozzles [1] cools down faster than the inner core UV resistant Sensitive to moisture effects then coated or embedded in polymers Filament diameter equals approximately 4 microns [4]. Stress-strain curves are linear elastic up to fracture and lose strength in case of bending or flexing	Load-carrying elements that provide: Tensile strength Dimensional stability (minimum elongation under external loads and environmental stresses Extension at the break Resistant to: Chemicals and microorganisms Effects of sunlight [3]
Finish layer as deposited solids on the glass fiber	Originates from an aqueous-based emulsion and consisting of adhesion promoter, protective polymeric film former, lubricants, surfactants, and an optional polymeric binder [5] Develops a strong interfacial bond between the polymeric matrix (coating) and the fiber by adhesion promoters [5] Forms the interphase region between the silane and resin (coating), as a semi-interpenetrating network consists of dissolved surfactants, monomeric and oligomeric silanols, [9] plus the diffusional length of the additives, which includes a film former [10] Expands the interphase region (in an acidic environment) into the glass fiber surface by the migration of silane into the empty spaces of the glass structure arising from solution of elements such as aluminum [11] Reaches an optimum performance of the interphase region by adjusting the yield strength of the interphase slightly lower than the yield strength of the surrounding media into which the fibers are introduced [12]	The finish layer establishes enough adhesion between coating and glass filaments in the composite structure for better stress transformation [4] Adhesion promotor (often a silane coupling agent [2]) provides handling, transport, strength protection, and compatibility with the coating matrix Film former imparts good handleability and controls wet-out kinetics [4] Lubricant aids the flow of the fibers through machinery without damage [5] Size coatings or binders prevent filament-to-filament abrasion [13] Coupling agents could also heal surface flaws and alleviate their negative effect on the fiber strength [14]
Primer, silicon	Is applied to the glass fabric by multi-pass dip-coating [15] in the form of aqueous dispersion [3]	Lubricates the glass yarns in order to improve flex and tear resistance [3] Provides a waterproof barrier [3]
Unfilled PTFE (polytetrafluoroethylene ethylene) coating	An opaque semi-crystalline thermoplastic that consists of linear or branched monomer units of tetrafluoroethylene [4,16] Composed of the twisting helix of a carbon-based core with an outer sheath of fluorine atoms, which protects the chain backbone (carbon-based core) from chemical attacks Good hydrophobic properties Is applied to the glass fabric by multi-pass dip-coating [15] in the form of aqueous dispersion [3] Dilution is adjusted in order to achieve optimum impregnation of fiber bundles and to fill in the openings of the woven fabric [3]	Provides a void-free surface to obtain enough adherence for the glass-filled coating layer [3]
Glass-filled PTFE (polytetrafluoroethylene ethylene) coating	For general PTFE features see above Is applied to the glass fabric by multi-pass dip-coating [15] in the form of aqueous dispersion [3] Very low surface energy and non-polarity, which makes it anti-adhesive and provides good self-cleaning and water-repellent properties [6] UV and IR radiation resistance [7] Relatively stiff [15] Incorporates hollow glass spheres (ballotini) with a diameter of 10–30 microns [3] (often extend above the coating surface [7]) to improve abrasion resistance and solar reflective properties (improving optical transmittance properties of opaque coating) to control the growth of fine cracks on PTFE surface Uses a hollow instead of a solid glass sphere because it formulates a more stable dispersion with a lower weight	Resistant to environmental impacts, chemical attacks, and mold growth [7] Reinforcement of glass filaments against humidity [3] Adds self-cleaning properties to glass-PTFE [3] Provides shear stiffness Stabilizes weave structures Controls the light transmission of the fabric [3]
Top coating	PTFE as the main coating is nonpolar, thus high-frequency welding is not possible [15] Melt processable fluoropolymer [3] such as FEP (fluoroethylene propylene copolymer), PEA (perfluoro alkoxy), or MFA (tetrafluoroethylene-perfluoro methyl vinyl ether) [7]	Increases the waterproofness, fungal attacks of glass-PTFE [6] Creates weldable surfaces Seals any surface imperfection [3]

**Table 2 materials-14-00846-t002:** Summary of investigations that assessed water impacts on glass-PTFE fabrics.

Researcher	Test Procedure	Results/Conclusions
Ansell et al. [4]	Several PTFE-coated glass fabrics with weights between 750 and 1600 g·m^−2^, manufactured by British, American, and German companies Immersion of specimens in a flask containing distilled water for 7 days in an environmental cabinet Conditions of 85 °C/85% RH (relative humidity) and 5 °C/low RH every 12 h Intermediate characterization after drying samples at 20 °C/65% RH	Retained tensile strength of about 62% of the virgin value Drying restored the stiffness and the reduction of the tensile strength is partially recovered Moisture plasticized PTFE, which yielded to greater degrees of yarn uncrimping and less fabric stiffness SEM micrographs indicated a poor bond between the ballotini and the PTFE matrix, which was susceptible to failure initiation when glass-PTFE was exposed to cyclic environmental or creep stresses
Toyoda et al. [39]	Non-coated glass fiber fabric and PTFE-coated glass fiber fabric Immersion at 20 °C, 30 °C, 50 °C, and 90 °C hot water bath for maximum of 14 days SEM method for assessing morphological changes	Changes in tensile strength of both materials depended upon the test duration and temperature By increasing temperature at each immersion time period, tensile strength decreased for both coated and non-coated specimens By increasing immersion time and temperature, water absorption went up for coated fabric Retained tensile strength at 90 °C after 14 days was 31% for uncoated and 62% for coated glass fiber fabrics At 20 °C, tensile strength did not change significantly for either coated or non-coated fabrics (maximum 5%) At 30 °C, the maximum tensile strength decrease was about 18% at 7 days of immersion for non-coated and 2% for coated at 14 days of immersion. After 14 days at 90 °C, micro-holes, micro-cracks, and partial failures on the surface of glass fibers appeared, whereas for the PTFE surface small indentations and adhesion of fine particles were observed
Asadi et al. [40]	Water tests at room temperature on glass-PTFE fabric type II for different time periods (24 h, 48 h, 72 h, and 144 h)	By stepwise increasing the soaking time, residual tensile strengths were about 85%, 91%, 94%, and 91% for warp and 82%, 89%, 86%, and 89% for weft directions

**Table 3 materials-14-00846-t003:** Specifications of the investigated material.

Sample	Number of Batches	Yarn	Total Weight ^2^ [g/m^2^]	Thickness ^3^ [mm]	Weave	Yarn Density ^4^ Warp/Fill [dtex]	Yarn Count Warp/Fill ^5^ [cm^−1^]
Glass-PTFE type II (III) ^1^	Two	E-glass fiber ^6^	1291	0.73	Plain 1/1	1360/1360	13/11
Glass-PTFE type III	Two		1153	0.66		2040/2040	11/13
Glass-PTFE type IV	Five		1641	0.94		4080/4080	8/10

^1^ Type II according to datasheet, type III according to measured tensile strength; ^2^ mean of all batches, measured based on DIN EN ISO 2286-2:2017-01 [42]; ^3^ mean of all batches, measured based on DIN EN ISO 2286-3:2017-01 [43]; ^4^ mass per unit length [0.1 g/km], taken from producers’ datasheets; ^5^ mean of all batches, measured based on DIN EN 1049-2:1994-02 [44]; ^6^ originally developed for insulators of electrical wiring, nowadays used as the reinforcement.

**Table 4 materials-14-00846-t004:** Residual tensile strength of glass-PTFE after different watering time periods.

Watering Hours	Residual Tensile Strength [%]	
	Warp	Weft
24	82.8 to 99.0	80.0 to 98.5
48	86.1 to 94.1	83.8 to 94.9
72	83.4 to 94.7	81.0 to 93.3
144	81.5 to 94.9	77.7 to 96.7
720	81.8 to 93.8	83.4 to 94.7

**Table 5 materials-14-00846-t005:** Magnitude of the tensile strength recovery and permanent residual tensile strength for glass-PTFE type II according to Table 3 after different watering time periods.

Watering Hours	Tensile Strength Recoveries [%]		Permanent Residual Tensile Strength [%]	
	Warp	Weft	Warp	Weft
24	+7.7	+9.0	92.2	91.3
48	+3.8	+1.0	92.6	89.9
72	−6.7	−0.7	86.9	85.6
144	+1.3	+4.6	92.5	93.6

**Table 6 materials-14-00846-t006:** Residual tensile strength of glass-PTFE for different water seepage mechanisms.

Water Seepage Mechanisms	Magnitude of Residual Tensile Strength [%]	
	Warp	Weft
In-plane + out-of-plane (two surfaces)	81.5 to 96.7	77.7 to 96.7
In-plane + out-of-plane (two surfaces) without surfactant	83.3 to 90.8	81.3 to 89.9
Out-of-plane (one surface)	83.1 to 95.3	86.0 to 89.6
Out-of-plane (two surfaces)	83.8 to 99.4	80.9 to 97.5
Out-of-plane (two surfaces) without surfactant wetting	84.3 to 92.1	85.2 to 93.8

**Table 7 materials-14-00846-t007:** Maximum strength modification factor k_hum_ for considering the influence of humidity for different investigated types of glass-PTFE samples according to Table 3 under different water seepage mechanisms.

Glass-PTFE Type II Warp/Weft	Glass-PTFE Type III Warp/Weft	Glass-PTFE Type IV Warp/Weft
1.20/1.25	1.12/1.15	1.25/1.20

## Data Availability

Data is contained within the article or Appendix A.

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
