# Peer review of "Water Influence on the Uniaxial Tensile Behavior of Polytetrafluoroethylene-Coated Glass Fiber Fabric"

_materials, 2021, doi:10.3390/ma14040846_

Round 1
Reviewer 1 Report
Thank you for submitting your work to materials. I have thoroughly reviewed the manuscript and feels it require significant improvement before it can be considered for publications. Please see my comments below:
- What about the novelty of the work? Add a specific section after the introduction to explain this.
- The introduction is very weak with minimum literature is presented. It is important to summarise the previous researches and highlight the gaps in the research. It should be thoroughly revised and a lot of information is required to be added in.
- To clarify, What does cf mean? (cf. Figure 2)
- What does section 2 represent? It is your work or literature? Authors are referring the citations everywhere but then linking with Table 2 which is shown way later in the manuscript. The presentation is very poor and the flow of the manuscript should be rechecked.
- If entire chapter 2 is literature, then merge it with the introduction to avoid the confusion.
- Usually, the measured tensile strength is higher than that given in the datasheet? What does it mean?
- Developed is not a correct word, replace with manufacturing (3.1.3)
- "Generally, these stress-strain curves include decrimping and yarn extension regions" Any justification, citation? What does decrimping mean?
- English and grammar should be significantly improved.
Please make all these changes and resubmit with point by point rebuttal to my comments.
Thanks
Reviewer 2 Report
This paper tested water diffusion into a glass-PTEE composite.
From the viewpoint of industrial materials and their test against water,
almost it should be accepted as it is.
However, from the viewpoint of basic science of each polymer compound,
Please mention or explain the relationship between behavior of water
of each layer (which has been already described in this paper closely)
and "molecular structure", for example, hydrophilic groups of polymers
or hydrophobic properties of glass etc. As this style, whole nature can
be understood phenomenologically, though the reasons of the results
are hard to understood or think for readers.
That's all.
Reviewer 3 Report
The Authors propose an experimental study on the influence of water diffusion and water-induced degradation for PTFE-coated glass fabrics used for building membranes for architectural structures. In particular, the tensile strength and stiffness reduction is studied for in-plane and out of plane watering, in different conditions. Also, the influence of time and the recovery after drying cycles is examined.
The paper is interesting, well written and organized, but I think that the minor revisions listed below could further improve it.
1) The introductory sections (1. Introduction and 2. Water seepage in architectural coated membrane fabric) could be enriched by referring to some details about the mechanics of PTFE-coated glass fabrics. These details could profitably support the considerations made in commenting on experimental results. Moreover, references are too few (8 + 2 manuals + 4 standards), and some of them are very old: the Authors are encouraged to give a wider and updated landscape of the literature: for example, papers on the mechanic of fiberglass fabrics, on the degradation due to watering, etc..
2) The novelty of the present paper with respect to the study in section 2.2 should be highlighted in the same section.
3) In section 3.1.3, some details on the measurement of the deformation should be introduced.
4) In section 4.1, I suggest adding a more precise definition of the secant stiffness employed and an explanation of why this particular secant stiffness is considered (a good idea could be to employ a figure).
Reviewer 4 Report
In the paper by Asadi et al., the effect of water on the mechanical properties of PTFE coated glass-fiber fabrics is studied. The authors performed a detailed, in-depth study with many results that provide useful insight into the mechanical deterioration of the fabrics. However, the article must be revised before its publication in Materials. Focus must be given on the novelty and significance of this work, to increase its potential target audience and impact. Moreover, the paper has to be restructured according to the guidelines of the journal.
Specific comments
- It is more common to use past tense in the abstract. It is not necessary to mention that literature was reviewed; short literature descriptions are always included in the introductions of original research articles.
- Introduction line 29: which membrane are the authors referring to? A particular one or coatings in general?
- Introduction: what are the disadvantages of PTFE coated glass-fiber fabrics and what are other fabrics that are being used for similar applications (e.g. PVC)?
- Introduction: what is the estimated life expectancy of PTFE coated glass-fiber fabrics? Are there reports on premature failure due to water?
- Introduction: after the recognition of the negative effects of moisture on PTFE coated glass-fiber fabrics, where there any suggestions or efforts to tune their durability?
- Introduction: it is common to include a paragraph in the end of introduction that describes the objectives of the work performed. In my opinion, this is not clear, nor the difference of this work with the ones mentioned in the introduction.
- Is the section 2 part of the introduction? If so, in my opinion it does not need to be in a separate section but can be merged with it.
- Materials: please mention the manufacturer of the fabrics if possible.
- Materials line 99: Does that mean that the manufacturer mislabeled their products on purpose?
- ISO 1421:2016 includes two methods. Which one was used?
- Figure 9: standard deviation is missing. One cannot safely comment on the trend of the stiffness without it. The same comment applies for figure 13.
- Figure 10: since no effect is concluded from these graphs they could potentially be moved to supplementary as the number of figures is already too big in the manuscript.
- Figures 15, 16: It is very difficult to interpret the result with these scatter graphs. The authors might want to consider presenting them in a different style of graph.
- Section 5 must be merged with the section if should belong to, i.e., results and discussion.
- Where does the calculated Maximum strength modification factor stand in comparison with other values reported, for example for other types of fabrics? Is a 20% reduction of tensile strength considered to be a lot?
- Do the authors believe this deterioration have visible effects on the everyday uses of these fabrics or is their tensile strength so big that even with its reduction they are still durable?
Finally, I would like to congratulate their authors for their work and thank them for taking the time to address my comments.
Reviewer 5 Report
This paper named “Water influence on the uniaxial tensile behaviour of polytetrafluoroethylene-coated glass fibre fabric” tested the impact of water on the tensile strength deterioration of different type of glass-PTFE fabrics. The authors neither analyzed the internal reasons nor put forward effective solutions. The integrity and novelty of this research paper are not enough for publication in Materials. Here are some suggestions: 1. Whether the microstructure of glass-PTFE fabrics has changed after exposure to water for different time and drying, the authors should provide the SEM contrast diagram. 2. Water definitely has influence on the tensile strength of glass-PTFE fabrics, why doesn't the impact change over watering time? 3. After drying, the stiffness of watering glass-PTFE fabrics will recover to a great extent. Why? 4. Figure 10 shows that the tensile strengths of different parts of the same sample varies greatly under the same watering conditions. Why does this happen? 5. The water used in the experiment is neutral, but the water in actual environment generally has certainly acidity and alkalinity. The authors are suggested to complete the relevant data. 6. From the article we know that water in its different forms will reduce the strength of glass-PTFE fabrics. So, how to improve it?Author Response
Please see the attachment.

Reviewer 6 Report
An overall well-designed and nicely written manuscript.
1. Please check minor syntax and grammar errors
2. Please cite recent references, if there are any
Round 2
Reviewer 1 Report
I am happy with the revision based on my comments. Thanks.
Reviewer 3 Report
The Authors revised the paper according to the reviewer’s comments. Now, in my opinion, the paper can be accepted for publication.
Reviewer 4 Report
The authors have revised the manuscript after considering all comments. I believe the paper can now be published as it is. I would like to thank them for taking the time to adress all of my comments.
Reviewer 5 Report
This paper named “Water influence on the uniaxial tensile behaviour of polytetrafluoroethylene-coated glass fibre fabric” tested the impact of water on the tensile strength deterioration of different type of glass-PTFE fabrics, which clarifies some disputes about the tensile strength decrease of glass fibres under water exposure. The structure and content of the article have become more complete after revision. I think it could be published in Materials.